# A Systematic Review of Social Sustainability Indicators for Water Use along the Agricultural Value Chain

Pascalina Matohlang Pilane *[ID], Henry Jordaan [ID] and Yonas T. Bahta [ID]

Department of Agricultural Economics, University of the Free State, Bloemfontein 9300, South Africa; jordaanh@ufs.ac.za (H.J.); bahtay@ufs.ac.za (Y.T.B.)
* Correspondence: mohlotsanemp@ufs.ac.za

**Abstract:** The concept of sustainable water use serves as an indicator of environmental, economic, and social pressure on freshwater resources globally; however, the social element of sustainability is not well researched within water-consumption studies. The objective of this paper is to consider the current state of the literature on social sustainability indicators for water use in agriculture, as well as to describe the social (people) element of sustainability and establish water use as an element of society. By combining viewpoints, systematic literature reviews address research topics with a strength that no single work can have. From 314 papers published between 2013 and 2023, 42 papers were eligible for the review. This work employed a mixed-methods approach that included a systematic review following the (PRISMA) framework, scientific mapping through VOSviewer software (version 1.6.19), thematic reviews, and a review of the grey literature retrieved from artificial intelligence and deep learning technologies. The findings indicate that social sustainability indicators are based on environmental indicators. There are no set standards for what to consider as a social indicator of water use or for how these indictors can be measured. Life-cycle assessment and water-footprint assessment frameworks have shown progress with indicators that capture the social value of water such as productivity-reducing externalities, equity, and jobs per cubic metre of water.

**Keywords:** sustainable freshwater use; social sustainability; literature review; Life Cycle Assessment; Water Footprint Assessment





## 1. Introduction

Scientific literature on the three pillars of sustainability has not given equal weight to the environmental (planet), economic (profit), and social (people) elements of sustainability studies, and the same is true of research on sustainable water use [1,2]. Sustainable development was first described by the Brundtland Commission in 1987 as "development that meets the needs of the present without compromising the ability of future generations to meet their own needs" [3]. The phrase "social sustainability" in itself conveys some of the complexity surrounding it, as there is no universally accepted definition of the term. Instead, social sustainability frequently depends on the context or on the source of the data being analysed when clarification is needed [4]. Due to difficulties in quantifying this social element, it is frequently omitted in favour of an environmental and economic approach, which leaves society (people) in a vulnerable state [4,5].

Vulnerability is defined as a situation caused by social, economic, or environmental elements or processes that increase a community's sensitivity to scarcity [6]. Subsequently, consensus within the scientific community shows that social sustainability focuses on personal indicators such as education, skills, experience, consumption, income, and employment, which are globally diverse and unique even on a provincial level, and all of which are related to a state of vulnerability. Nonetheless, the social value of water incorporated into agricultural products, which informs the social sustainability of water use, has not been given attention in scientific literature [1,2].

Agenda 2063 is the African Union (AU)'s development blueprint for Africa; it aims to achieve inclusivity and sustainable socio-economic development in the next 50 years. It includes seven targets, the first being to achieve a prosperous Africa based on inclusive growth and sustainable development [7]. The goal of this target is to eradicate poverty through social and economic transformation on the continent. This target will be realised through improved living standards, job creation, income equity, and sustainable management of the continent's land and water resources. There is an undeniable similarity to Sustainable Development Goal (SDG) 12 of the United Nations' (UN) Sustainable Development Agenda for 2023, which aims for sustainable production and consumption of products and services along value chains involved in food production [7,8]. Additionally, the 2012 National Development Plan (NDP) of South Africa aims to eradicate poverty and minimise inequality by 2030. This type of development cannot be achieved when the environmental, economic, and social elements of sustainability are not equally prioritised or understood.

Footprints serve as indicators of effects on the environment and provide a basis for comprehending environmental, economic, and social change, as well as the resultant impacts of these effects [9–11]. A water footprint serves as an indicator of human pressure on freshwater resources that can help researchers comprehend the environmental, economic, and social changes that result from water scarcity. However, the social water footprint is not realised or considered in water consumptive research; vulnerabilities and benefits that ordinary people derive from the water consumption of a product along agricultural food chains are therefore not fully understood in the academic literature.

Traditional approaches to gathering literature for reviews frequently fall short in terms of thoroughness and rigour and are often also undertaken on an ad hoc basis rather than in accordance with a predetermined methodology [12]. This approach raises concerns regarding the validity and reliability of such reviews. Thorough evaluations of the literature, particularly systematic reviews, can address research issues with a strength that no one paper has by integrating information that would otherwise be omitted [12,13]. Additionally, a systematic literature review generates evidence on a meta level and identifies areas where additional research is required. Scientific mapping makes it possible to present data graphically through category maps that highlight network linkages within the academic literature [14,15], while thematic reviews offer a closer look at the direction in which the literature leads, as discovered through an extensive literature search.

The objective of this paper is to consider the current state of the literature on social sustainability indicators for water use in agriculture. To assemble the available literature on the social element of sustainability, the authors of this paper conducted a bibliographic review of social sustainability indicators for freshwater use along agricultural value chains. For 314 papers published between 2013 and 2023, the research methods included a systematic review following the Preferred Reporting Items for Systematic Review and Meta-Analysis (PRISMA) framework, scientific mapping through Vosviewer software, thematic reviews, and artificial intelligence (AI) to filter the grey literature. This paper offers an overview of the state of the literature concerning social sustainability indicators for agricultural water use to establish water as an element of society. The review further highlights the lack of and need for social-sustainability valuation of freshwater resources along agricultural value chains.

The paper is structured as follows: Section 2 describes the methods used to conduct the review. Section 3 includes the results, in the form of scientific mapping, on social aspects of sustainable water footprint literature, a thematic review of social sustainability along agricultural systems and a description of grey literature on established water-use frameworks that consider society, as retrieved from an AI software (OpenAI 2015–2024) program. Lastly, the results include a thematic review of the inclusion of social factors in water-footprint-related research methodologies. Section 4, the discussion, establishes the social aspects of water use. Finally, Section 5 discusses the study's conclusions and limitations, as well as the need for further research.

## 2. Methods

This review included a mixture of methods used to review the literature, including the accepted protocol for reporting evidence in systematic reviews and meta-analyses (PRISMA). VOSviewer visualisation software was used for scientific mapping. Following the bibliographic networks are three thematic reviews that were conducted with additional literature to explore the revealed themes. The first theme focuses on social sustainability indicators for agricultural production value chains. The second theme focuses on social sustainability frameworks for freshwater resource use derived from grey literature through AI generated software. The last theme explores social sustainability assessment within water footprint methodologies. The paper also conducts a traditional literature review on the social character of water.

A mixed-methods approach facilitated an extensive review of the literature that is not limited to scientific publications but also included the grey literature. This method accommodates different elements of the literature such as a combination of keywords and network linkages derived from a systematic review of the literature, as well as thematic reviews that allow a broader discussion of related themes within a specified research question.

The research design is depicted in Figure 1, which presents the research problem, research question, methods, interpretation of the literature, and summary. Included in the review were 314 documents retrieved from Scopus. The search query is shown in Table S1, and the inclusion/exclusion criteria is explained in Table S2. Tables S1 and S2, are found in the Supplementary Materials of this review as Appendices A and B. To corroborate and better discuss the information derived from the systematic review process, additional literature was retrieved and is discussed in the thematic reviews.

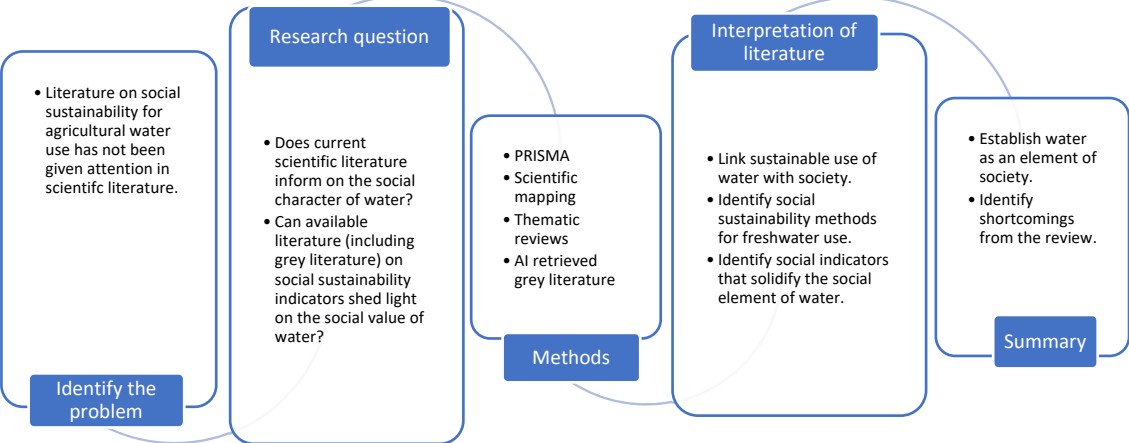

**Figure 1.** Research design. Source: Authors.

### 2.1. Prisma Method

PRISMA permits the replication of review methods because of the transparency of this process [16]. To begin a systematic review of the literature, 314 papers were selected from the results from the query string depicted in Table S1. From the collected papers, duplicates were removed, after which 253 documents remained. After the papers were screened based on their title, abstract, and content, 30 journal articles and 12 papers from the grey literature remained, for a total of 42 papers. The PRISMA flow diagram is depicted in Figure 2.

### 2.2. Scientific Mapping

Vosviewer, which was created by Van Eck and Waltman [17], uses visual features based on mapping techniques to turn Research Information Systems (RIS) file formats into clusters and diagrams, permitting researchers to analyse information such as authors, citations, co-citations, and keywords [18].

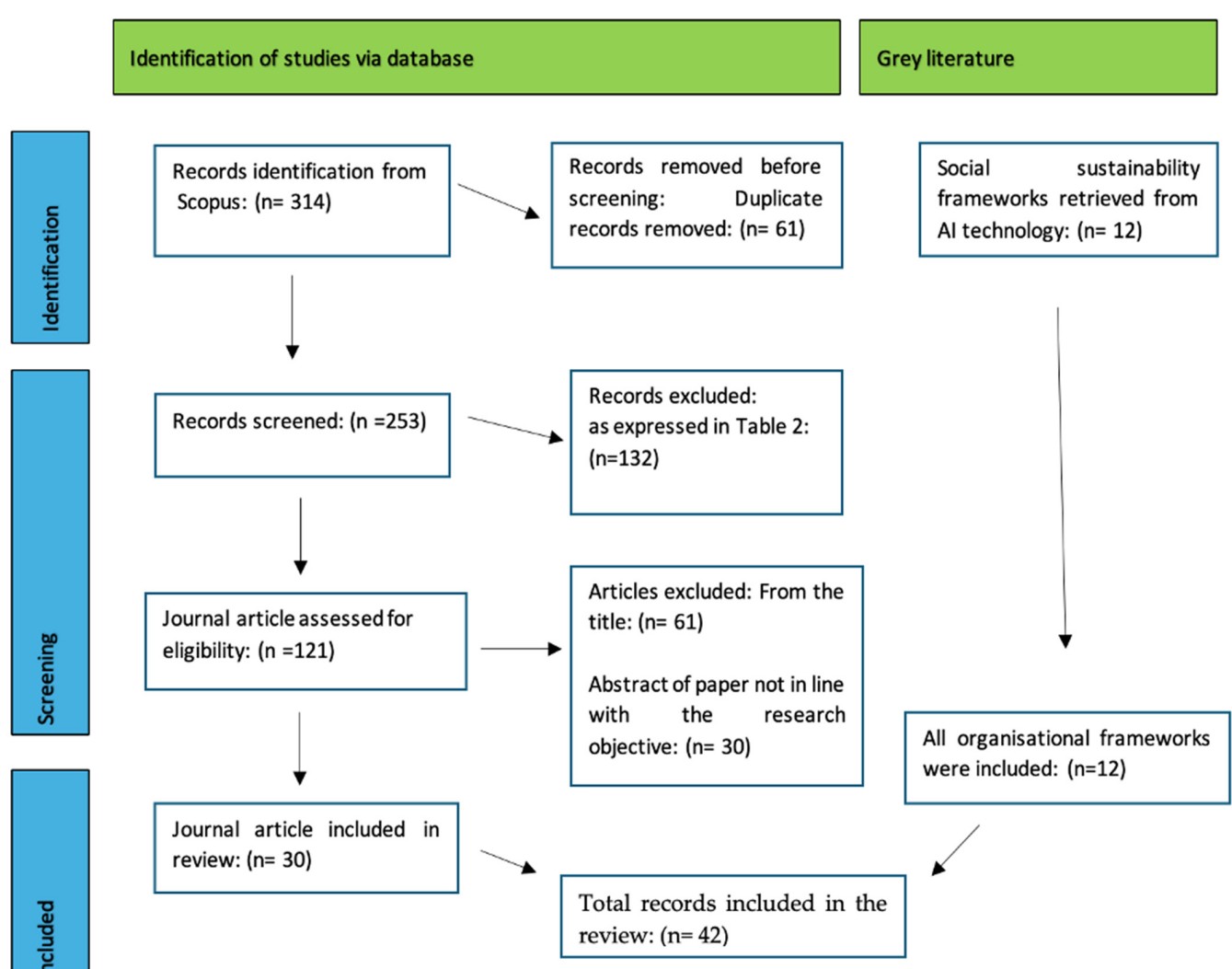

**Figure 2.** PRISMA schematic representation of the inclusion/exclusion criteria. Source. Author's adoption from PRISMA [16].

From the papers shown in Figure 2, 30 scientific papers were eligible for inclusion in this review. These papers were further analysed through the VOSviewer visualisation software to display bibliometric networks, and the analysis considered co-authorship and co-occurrence of keywords within the specified search query to elucidate the key concepts on social elements of sustainable water-use research.

*2.3. Thematic Review*

Thematic review methodology is generally characterized as a strategy for detecting patterns in the form of themes to identify components of an idea within a research area [12,19].

The thematic review on social aspects of sustainable water-use frameworks was conducted with the use of grey literature, which was retrieved through artificial intelligence technology. In particular, the researchers employed Chatbot Generative Pre-Trained Transformer (ChatGPT) from OpenAI, which mimics conversations with human users. The choice to use AI technology was made to diversify the literature-search methods and thus to access a pool of literature not included in published literature reviews. ChatGPT chatbot operates by using algorithms that are programmed to understand natural language inputs and respond with relevant, pre-written responses [20,21]. ChatGPT is continually devel-

oped with reinforcement techniques, natural language processing, and machine learning to better comprehend and respond to users' demands [20,21].

For academic writing, ChatGPT is helpful in identifying research questions, providing an overview of the current state of a subject, and assisting with tasks such as formatting and language review. For this paper, ChatGPT was used to obtain grey literature on established water-use frameworks that consider the social element of sustainable water use, as well as established indicators that consider the social element of sustainability. All the responses were verified, and relevant sites were further reviewed. Where clarity was needed, additional literature on the tool was collected from scientific resources.

## 3. Results

### 3.1. Scientific Mapping

Network mapping presents the network linkages derived from the search query detailed in Table S1 and the papers selected from the PRISMA framework, as shown in Figure 2. Figure 3 represents the co-authorship network, and Figure 4 represents the co-occurrence linkages.

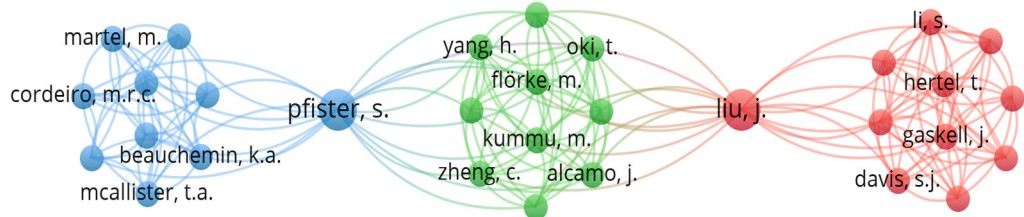

**Figure 3.** Co-authorship linkages from the literature on social aspects of sustainable water footprints. Source: Authors.

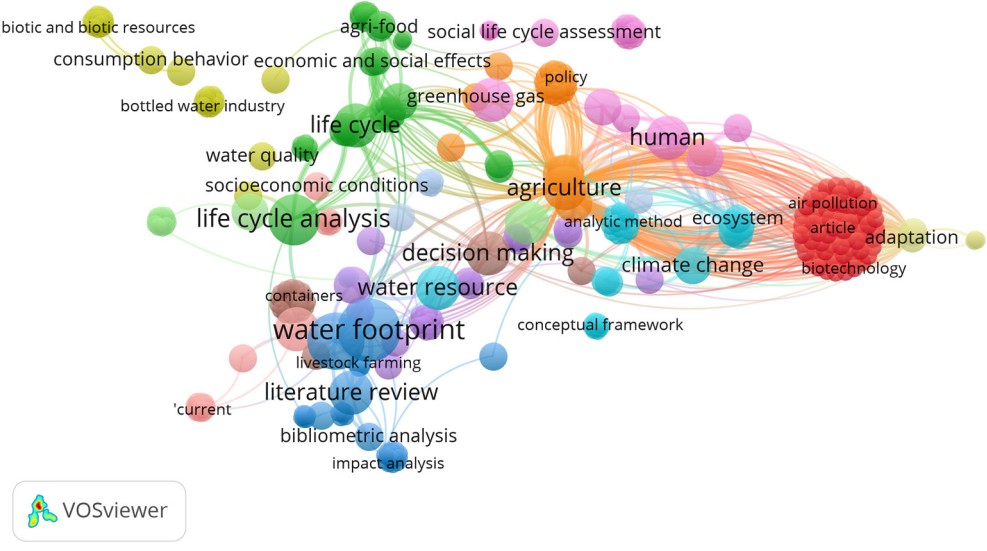

**Figure 4.** Co-occurrence linkages for the literature on social aspects of sustainable water footprints. Source: Authors.

### 3.1.1. Co-Authorship Network Linkages

Following the logic of Contreras and Abid [14], Yang and Thoo [15], and Cavalcante et al. [22], who detailed the value of bibliometric visualisation software, VOSviewer was used to describe network linkages among co-authors, as well as the co-occurrences between keywords. Of the three hundred and fourteen papers screened from Scopus, thirty were qualifying papers, as depicted in Figure 2. These thirty papers involved one hundred and thirty-seven authors; the minimum number of documents per author was set at one, and

the total strength of the co-authorship links was calculated. From these calculations, the largest set of connections consisted of 31 authors and 166 links.

Figure 3 depicts the co-authorship relationships among the selected papers. which shows three clusters connected by authors Pfister and Liu. The first cluster is strongly based on a board-invited review titled "Quantifying water use in ruminant production", which was published in the *Journal of Animal Science* and cited 47 (the number of citations was updated upon first submission of this review article) times. The second cluster is based on an American Geophysical Union publication in the journal *Earth's Future*, titled "Water scarcity assessment in the past, present and future". This paper was cited 770[1] times. The third cluster, published in the *Science* journal, is titled "Systems integration for global sustainability", and had 1192[1] citations. The number of citations was updated upon first submission of this review article. An in-depth analysis of the three papers that form the above clusters led to the below conclusions.

### Board-Invited Review: Quantifying Water Use in Ruminant Production [23]

The authors stated that water-consumption approaches such as the Water Footprint Assessment (WFA), Life Cycle Assessment (LCA), and Livestock Water Productivity were developed based on the realisation that water is a finite good. Legesse et al. [23] conceded that these methods differ with respect to their target outcome, geographical focus, description of water sources, handling of water-quality concerns, and interpretation and communication of results. This board assessment brought together methodological pioneers such as Hoekstra, who conceptualised the WFA tool, and Pfister, who conceived the LCA tool; both authors measure the sustainability of freshwater use.

### Assessment of Water Scarcity in the Past, Present, and Future [24]

The authors examined a range of indicators designed to capture various aspects of water scarcity, which is considered a major constraint to socio-economic development. The main components of these indicators were population, availability of water, and water use. Based on their findings, Liu et al. [24] provided an overview of the progress in water-scarcity assessment wherein they indicated that the WFA and LCA have improved as sustainability indicators, as has the Intergraded Water Quantity-Quality Environmental Flow Indicator developed by Zeng et al. [25]. The authors concluded that despite the progress made, the abovementioned indicators struggled to incorporate the complexities of socio-economic water demand and that interactive indicators within these tools were necessary.

### Systems Integration of Global Sustainability [26]

Seeking system-based or holistic approaches that integrate human and natural solutions, this paper advances the need to consider socio-economic and environmental effects simultaneously, rather than separately. Liu et al. [26] highlighted the need for new and inclusive indicators that incorporate social, economic, and ecological uses of freshwater resources.

### 3.1.2. Keyword Co-Occurrence Linkages

Figure 4 shows the co-occurrence analysis of keywords and the linkages between the keywords of the 30 scientific papers retrieved from Scopus. The results include 444 keywords, 15 clusters, and 10,159 links between these words. The most prominent cluster included keywords such as "air pollution", "acidification", "carbon dioxide", "health", and "ecosystem", which represent indicators closely related to the environment and which also serve as health indicators that impact society. This cluster was followed in significance by smaller clusters including keywords such as "agriculture", "environmental policy", and "sustainability". "Life cycle assessment" appeared three times in separate clusters, and "water footprint" was included among the most prominent clusters; the latter was expected, considering the search query. Words related to the social element of sustainability included a relatively small cluster with the keyword "economic and social effects".

Figure 5a–d represents an in-depth analysis of the keywords from the co-occurrence linkages indicated in Figure 3. Figure 5a illustrates strong linkages between the terms "water management", "water footprint", "bibliometric analysis", and "ecological indicators". This cluster is followed in size by a cluster that illustrates linkages between "absolute sustainability" and "risk assessment", "land use", "water pollution", and "consumption-based accounting". Figure 5b illustrates less dominant clusters wherein "life cycle" is linked to economic and social factors such as "agri-food" and "employment", as well as to "production" and "consumption". This is followed by a cluster linking the SDGs to "consumption behaviour", "industrial production", and "global community". Figure 5c depicts strong links between "agriculture", "carbon footprint", "environmental sustainability", "greenhouse gas", "social sustainability", "government", and "socioeconomics". Lastly, the dominant cluster highlighted in Figure 5d includes keywords associated with the environmental element of sustainability. These words include "carbon dioxide", "air pollution", "environmental change", "atmospheric pressure", and "nitrogen", as well as "dietary", which is mostly associated with indicators such as "overnutrition" and "physical activity". Societal indicators such as "food security" and "population growth" are found in the same cluster.

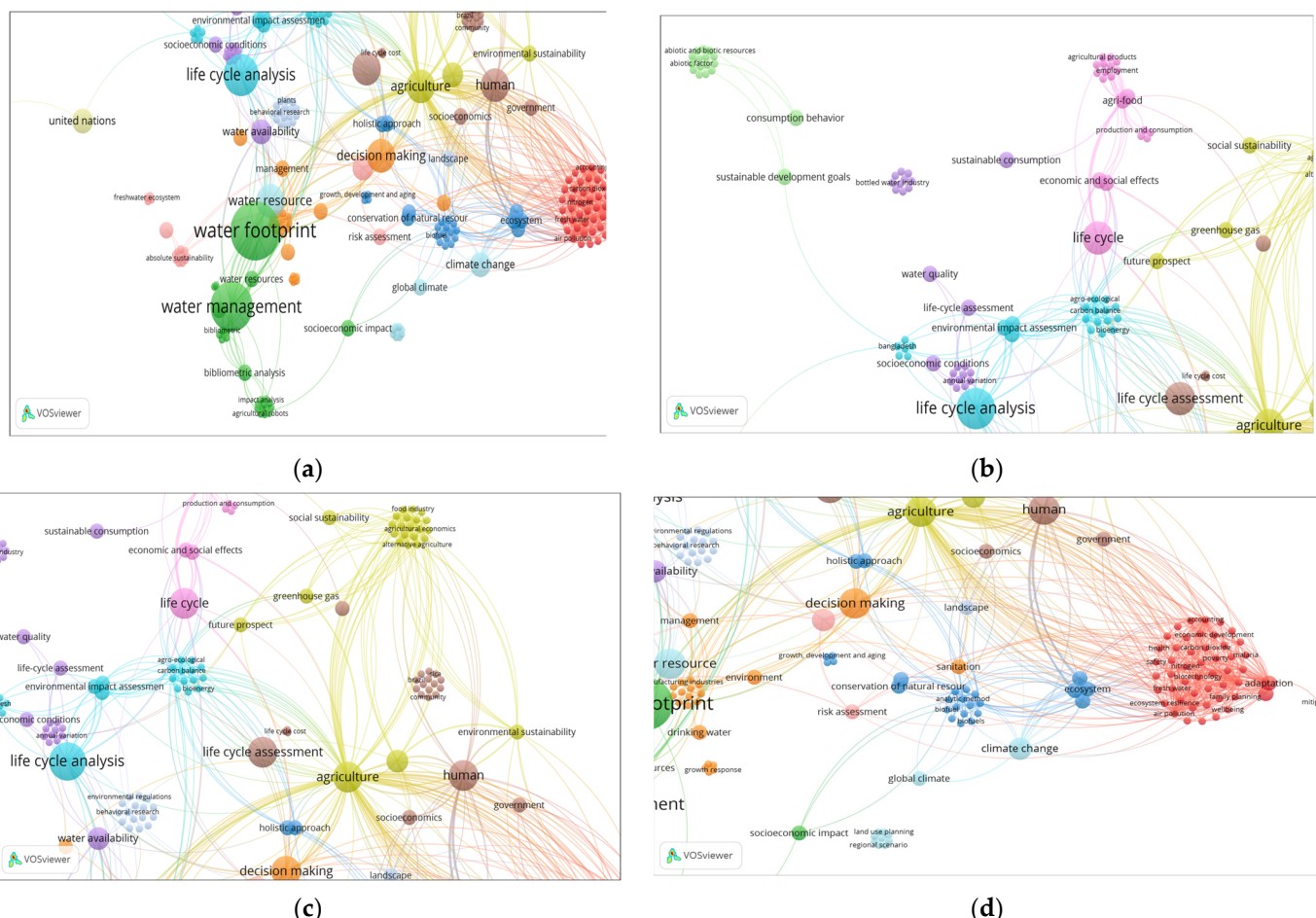

**Figure 5.** Detailed depiction of the co-occurrence network illustrated in Figure 3. Source: Authors.

Next, this paper includes a thematic review of the literature that considers social sustainability tools and indicators for agricultural production value chains to identify dominant social indicators for sustainable agriculture.

*3.2. Thematic Review of Social Sustainability Indicators for Agricultural Production Value Chains*

Sustainability indicators within tools are intended as specific, observable, and measurable characteristics that can be used to demonstrate a change or progress towards a specific outcome [1,2,27]. To comprehend the social indicators for water use incorporated into food production, it is necessary to explore social sustainability tools in the context of agricultural value chains. Although these tools do not measure social vulnerability in response to freshwater scarcity, their value lies in identifying the role that social indicators play in agricultural food chains.

Janker and Mann [27] evaluated indicators for 34 social sustainability tools. The selection criteria included the name of the social dimension, a definition of what the social dimension should entail, the underlying concepts and frameworks, how specific indicators were implemented, and the topics addressed by the indicator sets. The authors indicated inconsistencies where social sustainability tools were either based on human rights or working rights as per the UN and international labour organisations or on farmers' perceptions of quality of life, with recurring topics such as labour conditions and societal impact. The authors concluded that there was a lack of consensus on the social dimension of sustainability and that the scope and standard should be context-dependent in a locally embedded manner [27]. Desiderio et al. [2] reported 34 social sustainability tools based on five food supply chains and four stakeholder groups via an approach adapted from the Social Life Cycle Assessment (S-LCA) guidelines. The authors noted six stakeholders, namely the farmer, society, the worker, labour, consumers, and value-chain actors, as the main stakeholders in the social element of sustainability. The dominant stakeholders, their respective indicators, and the descriptions of each indicator are given in Table 1 The authors concluded that the number of industry and governance tools that measure social sustainability indicators has increased, while academic literature lagged [2]. Zhang et al. [28] considered social sustainability indicators from the Sustainable Agriculture Matrix tool, an approach that aligns with the official indicator for sustainable agriculture (SDG 2.4.1). These social indicators included resilience, health and nutrition, farmers' wellbeing, equity, and farmers' rights, as depicted in Table 1. Sannou et al. [29] investigated 36 indicators following the structure proposed by the Food and Agriculture Organization's (FAO) Sustainability Assessment of Food and Agriculture (SAFA) and Rural Institutions, Services and Empowerment (RISE) framework to integrate the social perspective into sustainability assessments in agricultural food systems. The authors included eight themes, namely (1) food security; (2) healthy and safe food products; (3) farmers' health and safety; (4) labour and working conditions; (5) decent livelihoods; (6) farmers' training; (7) social cohesion, security and conflict; and (8) land and property rights. These indicators are further explored in Table 1.

As shown in Table 1, social indicators for social sustainability along agricultural production value chains are widespread. Employment is a common thread that may be related to wages, education, or employee characteristics such as gender and education. Indicators are positioned as either part of the production stages or as part of health and labour relations, all of which involve the social impact of natural resources embedded in agricultural products because these too have an impact on employment along each stage of production. In summary, the authors reiterate that there is still a desperate need for empirical case studies of how aspects of social sustainability can be measured [29]. The following section focuses on tools from outside the scientific literature that evaluate the social value of water.

*3.3. Thematic Review of Frameworks That Evaluate the Social Aspects of Sustainable Use of Freshwater Resources Derived from AI Software Technology*

For the search of the grey literature, the following terms were typed into the ChatGPT chatbox: 1. "Social sustainability indicators for Water Footprint Analysis with references" and 2. "Social sustainability indicators within tools for water footprint with references". These prompts resulted in the following tools/frameworks and individual indicators.

**Table 1.** Summary of social sustainability indicators.

| | Major Aspect | Stakeholder | Indicator | Indicator Description |
|---|---|---|---|---|
| Desiderio et al. [2] | Production stage | Farmer | Health | n * of injuries<br>n * of fatality rates<br>Occupational illness |
| | | | Employment | % of total employment<br>-Male<br>-Female |
| | | | Labour | % of women working in agriculture<br>n * of women employed in the company |
| | | | Freedom of association | - |
| | Processing stage | Worker | Fair, equal, and healthy working conditions | Living wage per month<br>Min. wage per month<br>Ave. wage per month<br>n * of worker associations<br>n * of trade unions<br>Psychological support structure |
| | Wholesale stage | Worker<br>Society<br>Consumer<br>Farmer | Employee quality of work | Employee image<br>Employee expectations<br>Value perceived by employee<br>Employee satisfaction |
| | Retail stage | Worker [1]*<br>Society [2]<br>Consumer [3] | Supplier standard [2]<br>Responsible supply chain [3]<br>Internationally recognised responsible preproduction standard [4] | -<br>-<br>- |
| | Consumer | Consumer [1]<br>Society [2] | Consumer company identification [1]<br>Customer loyalty [2]<br>Reputation of the company [3]<br>Credibility of the company [3] | - |
| Zhang et al. [28] | Resilience | - | Crop production diversity | H Index crop diversity |
| | | - | Food affordability | Food affordability by low-income population |
| | Health and nutrition | - | Undernourishment | - |
| | Farmers' wellbeing | - | Rural poverty ratio | - |
| | Equality | - | Global gender gap | Report score |
| | Farmers' rights | - | Land rights | |
| Sannou et al. [29] | Food security | - | Nutritional need/dietary diversity | Experience of hunger<br>Quantity of home-grown food for consumption |
| | Healthy and safe food products | - | Safety/quality of food products | Nutritional security of produced food<br>Products and compatibility with set standards |
| | Farmers' health and safety | - | Access to healthcare | Access to safety and health training |
| | Labour and working conditions | - | Contribution to job creation | Employment provided by agricultural sector |
| | Decent livelihoods | - | Equity | Source of labour for rural populations<br>Gender and social equity<br>Capability of future generations to live sustainably |
| | Farmers' training | - | Farmers' education level | Changes in farm-management practices<br>Awareness of environmental protection |
| | Land and property rights | - | Land tenure | Rights of ownership and use of land |

*, [1, 2, 3,] and [4] denote indicator corresponding to stakeholder. Source: Authors' own analysis.

### 3.3.1. Tools

#### The Water Footprint Network's (WFN) Water-Stewardship Standard

Although it is listed as a water-stewardship standard that includes a focus on social responsibility and emphasises the need for assessing and addressing the social impact of water management, "water stewardship" appears only twice in the WFA framework by Hoekstra et al. [11]. The first appearance is in the acknowledgement of the Alliance for Water Stewardship (AWS), and the second is in a reference to the Coca-Cola Company and The Nature Conservancy. The WFN's Water Footprint Framework is discussed further in Section 3.4.2.

#### Water-Scarcity Atlas (Water Scarcity Footprint [WSF])

Strongly related to water footprint, the WSF evaluates water scarcity associated with the production of various crops. It includes indicators related to social impacts, such as the potential impacts of water use on local communities and livelihoods. The WSF is discussed in more detail in section Water Stress Footprint (WSF) this paper.

#### The Social Life Cycle Assessment (S-LCA)

The S-LCA is a framework that is designed to assess the social impacts of products or processes throughout their life cycle. Although not specific to WFAs, it can be adapted to evaluate social aspects within such assessments. The S-LCA includes indicators related to labour conditions, human rights, community engagement, and health and safety.

The S-LCA is discussed in more detail in section Social Footprint due to its close relationship to the LCA.

#### The Global Water Initiative's (GWI) Social Impact Assessment Tool

The GWI offers a practical field-based research guide that measures the impact of water-management interventions, including social aspects. Varady et al. [30] state that although the Millennium Developmental Goals (MDGs) and SDGs have enhanced the relevance of United Nations Water (UN-Water) and the network of water-related UN agencies, these efforts have not duplicated the efforts of the GWI but rather diversified its relevance and interrelatedness. The GWI's activities raise awareness and promote the sharing of information and the building of networks, while the GWI's behaviour reflects and influences water governance [31].

#### Alliance for Water Stewardship (AWS) Standards

The AWS contributes to the sustainability of local water resources with the inclusion of social sustainability considerations for responsible water use. With three certification levels—AWS Core, Gold, and Platinum—the multi-stakeholder approach addresses the social dimensions of water stewardship and promotes a holistic and sustainable water-management path [32]. AWS actions are undertaken in five steps, namely: (1) gather and understand, (2) commit and plan, (3) implement, (4) evaluate, and (5) communicate and disclose. Guidance on catchments, important water-related and stakeholder engagement are included as special subjects.

#### Global Reporting Initiative (GRI) Standards

The GRI standards offer a thorough structure for reporting on sustainability. Indicators of water management, such as social factors, including community involvement, water accessibility, and the effects of water use on nearby populations, are included in these guidelines [33]. The reporting initiative includes individual topics, such as water and effluent (topic 13.7), local communities (topic 13.12), food security (topic 13.9), employment practices (topic 13.20), and living income and living wages (topic 13.21) under GRI 13: Agriculture, Aquaculture, and the Fishing Sector 2022 [31,32].

Table 2 displays the social indicators of the AWS standards, as well as those of the GRI, and shows how these indicators are measured.

**Table 2.** Guidelines, indicators, and measures of the GRI and AWS.

| Organisation | Guideline | Indicator | Measure |
|---|---|---|---|
| GRI | Contextual disclosure | Water use in specified region | - |
| | Stakeholder engagement | Engage stakeholders in water use strategies | Local community |
| | Water governance | Disclose related risks and opportunities | Policies and practices |
| | Water monitoring and management | Related impact | Disclose water consumption, withdrawal, and discharge |
| | Water quality | Related impact | Impact on ecosystems and communities |
| | Compliance | Laws and regulations | Incidents of compliance/noncompliance |
| | Collaboration | Collaborative efforts with stakeholders | Collective response actions |
| AWS | Good water governance | Contributions to good catchments | Number and nature of contribution |
| | Sustainable water balance | Total volumetric water use | Measure of change |
| | Good water quality | Quality of effluent and receiving water body | Number of improvements |
| | Important water-related areas (IWRAs) | Identified IWRAs | Number and hectares |
| | Safe water for all | Hygiene awareness and access for stakeholders | Number of certified sites |

Source: AWS [30]; GRI [31].

As shown in Table 2, there again seems to be no consistency regarding water management tools, as there are no standardised means of measuring these tools. The results of the search for established indicators that considered the social element of water use follow.

### 3.3.2. Indicators

Established indicators for social sustainability from water-related tools are listed and described in Table 3.

**Table 3.** Social sustainability indicators for sustainable use of freshwater resources.

| Indicator | Indicator Description |
|---|---|
| Access to clean water | Access to, availability of, and quality of drinking water for communities |
| Sanitation services | Evaluation of sanitation services and wastewater treatment |
| Health outcome | Water-related diseases, child mortality rate, and waterborne diseases |

**Table 3.** *Cont.*

| Indicator | Indicator Description |
|---|---|
| Gender equity | Assessment of gender-distributed access to water and sanitation services and water-related tasks (United Nations Entity for Gender Equality and the Empowerment of Women |
| Livelihoods and employment | Impact of water on local livelihoods and employment opportunities |
| Community engagement | Effectiveness of local water governance and extent of community involvement |
| Cultural heritage and values | Effects of water use on cultural heritage and traditions |
| Social equity | Assessment of the distribution of water-related benefits.Impact on marginalised and vulnerable groups |
| Resettlement and displacement | Evaluation of the potential for resettlement and impact on affected communities |
| Conflict and social tensions | Tracing incidents of water-related conflicts |
| Local economic development | Assessing the contribution of water use to<br>- local economic development<br>- income generation<br>- poverty reduction |
| Food security | Impact of water use on food production, availability, and affordability |
| Education and awareness | Awareness and education on sustainable water use within communities (United Nations Educational, Scientific and Cultural Organization |
| Social resilience | The ability of a community to adapt to water-related challenges |
| Social satisfaction and wellbeing | Overall wellbeing and satisfaction of the local population regarding water management |

Results from Tables 2 and 3 suggests that social sustainability tools and indicators are viewed in isolation, which creates a challenge in attempting to account for the social character of water; this result was expected, considering the complexities of an unstandardised method of analysis. To narrow this search in relation to social aspects of sustainable freshwater resources from the previous sections, water-footprint methodologies emerged as possible methods of assessment. It is for this reason that the following section considers social sustainability assessment within water-footprint methodologies.

*3.4. A Thematic Review of Social Sustainability Assessment within Water Footprint Methodologies*

The International Organization for Standardization's (ISO) standard 14046:2014 serves as a guideline for what should be included in a thorough water-footprint study [34]. It states that the term "water footprint" can be used only to characterise the outcome of a thorough impact assessment [34], whereas this type of assessment is a measure of the potential effects of water on the environment, economy, and society. However, research on ISO 14046 paid little attention to the social indicator within water-footprint studies; it has been repeatedly noticed that a thorough investigation of this dimension of sustainability is frequently foregone in such studies in favour of environmental and economic analyses [35–37]. With this said, the LCA and WFA were identified as comprehensive freshwater sustainability assessment tools that monitor and evaluate sustainable consumption and production patterns at a local, regional, or global level [35]. This section addresses the social element of sustainability within the LCA and WFA frameworks.

### 3.4.1. LCA

The LCA addresses aspects and potential environmental impacts throughout a product's life cycle, in compliance with ISO 14046:2014. This assessment involves four phases: (1) the definition of the goals and scope of water-footprint analysis, (2) water-footprint inventory analysis, (3) water-footprint impact assessment, and (4) water-footprint interpretation. Although it has been acclaimed as a comprehensive WFA, the LCA does not address the social or economic impact of a product's water footprint. These aspects are considered within the S-LCA outside the scope of water footprint. Nonetheless, comprehension of the S-LCA, as well as the social footprint, beings researchers a step closer to understanding the types of social indicators and how they are measured along agricultural supply chains—an understanding that will be meaningful when analysing the inclusion of social factors in water-footprint studies. This highlights the missing link between freshwater consumption for production and the social sustainability of this finite resource.

### S-LCA Framework

According to [38,39], the S-LCA is defined as a technique for gathering, analysing, and communicating information about the social conditions and impact of production and consumption. The primary objective of the S-LCA is to provide decision-making support after analysis of changes in the lives of workers, consumers, society, and other key stakeholders associated with the life cycle of a product. The S-LCA framework considers the social impact of products and processes throughout their life cycles. The S-LCA is inclusive of indicators strongly related to human rights, community engagement, labour conditions, health, and safety. Although not specific to water footprint, the S-LCA can be of great value in recognising important social indicators and how they are used and can be adapted to articulate social aspects of sustainable production and consumption. Two main impact assessments, namely the reference scale approach (Type I) and the impact pathway approach (Type II), relate to the social footprint.

### Social Footprint

According to [40,41], social footprint is a comprehensive monetary measure that assesses income redistribution and the cumulative impact of productivity-reducing externalities associated with a specific product or activity. It consists of the following three main components:

### Income Redistribution

This component measures the overall gain or loss of societal value that comes from the distribution of wealth among various social groups.

- Transferring funds from wealthy consumers to underprivileged farmers, for example, is viewed as enhancing the general usefulness or wellbeing of society.

### Productivity-Reducing Externalities

These externalities include the social effects of aspects such as poor infrastructure, corrupt officials, and weak governance. The calculation entails determining the difference between the actual value added from a work activity and the potential value added.

- The latter is calculated using a specified economy's current value added per work hour after accounting for factors that currently reduce productivity.

### Monetised Social Benefits

In addition to income redistribution and productivity-reducing externalities, social footprint considers monetised social benefits that arise from positive actions taken by companies in the supply chain.

- These actions, categorised as "creating shared value", contribute positively to the overall social impact.

To correctly account for LCA approaches to social sustainability, it is important to consider the WSF.

Water Stress Footprint (WSF)

Developed by Pfister et al. [42], the WSF assesses water shortages related to the cultivation of different crops. It contains social-impact indicators such as the possible effects of water use on nearby people and means of subsistence. Sturla et al. [43] conceptualised a further measure of footprint, namely the Social Scarce Water Footprint (SoSWF), to account for social factors that affect the availability of water. These authors argue that water footprint should address the environmental impact generated by the exploitation of water resources, such as Scarcity-weighted Water Footprint (SWF) [42,43]. The SoSWF is based on the concept of the Scarce Water Stress Index (SWSI), which is an indicator of per capita water availability [42,43]. It is further argued that the SWF delivers unrealistic sustainability ratings because the single indicator of blue water stress and blue water efficiency undermines sustainable production [44].

3.4.2. The WFA

The WFA is a tool that promotes water productivity and sustainability of water use and simultaneously enhances the management of water resources around the globe [9,10,35,45]. Water-footprint sustainability assessment is inclusive of environmental, economic, and social indicators of wise water allocation [9,35,45]. The sustainability assessment method consists of four steps, as indicated in Figure 6, namely: (1) identification of sustainability criteria, (2) identification of hotspots, (3) quantification of the primary impact, and (4) quantification of secondary impact. As stipulated in the water-footprint manual, only when the goal or scope of an analysis goes beyond hotspot identification, in other words, when it encompasses environmental, economic, or social violations of freshwater use, is it deemed necessary to generate a detailed description of how water footprint impacts sustainability indicators [9]. The WFA framework therefore does not necessarily evaluate progress towards desired sustainability outcomes in the absence of water-use violations, an omission that hinders progress towards achieving responsible consumption and production initiatives such as Agenda 2063. Furthermore, literature on WFAs typically includes only an environmental sustainability analysis, ignoring the economic and social sustainability criteria required for a comprehensive WFA.

The WFA framework for social sustainability is loosely interchanged, and incorrectly so, with social equity, which is defined as a measure of the evenness of water consumption [9,46]. Hoekstra and Wiedmann [9] reiterated that the social indicator of sustainability for a WFA is consumption behaviour that translates into environmental footprint. This depiction of an indicator that collectively considers the impact on individuals and populations who form societies cannot provide a sufficient analysis of a multifaceted resource such as water.

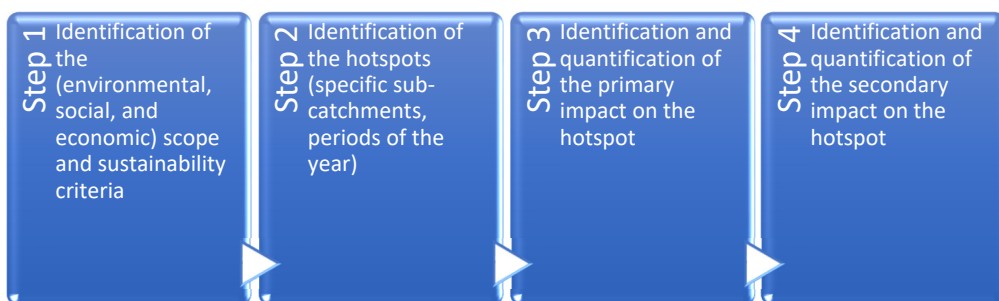

**Figure 6.** Assessment of the sustainability of water footprint in a catchment. Source: Hoekstra et al. [11].

In-depth analysis of the social indicators of sustainability according to the WFA includes human health, employment, distribution of welfare, and food security [9,35].

Following the categorisation theories described by Sannou et al. [29], the social indicators for water footprint were categorised by their definition;if the definition was not fully understood, categorisation was performed by looking at similarities between concepts. As a last attempt to categorise indicators, the differences between neighbouring categories (decision-bound theory) were considered.

The WFA does not consider the opportunity costs of water, which cripples its ability to allocate already scarce water resources. Social equity, portrayed as water footprint per capita, does not provide a true indication of the social value or impact of freshwater consumption along a product's value chain; therefore, one cannot use this metric to understand the effect on the lives of those living within these catchments [44,46]. Logically, the authors hold the position that a thorough analysis in terms of value added to society within the WFA would equip water managers to compare production systems in terms of the gains and losses observed by society [47–49]. Table 4 shows the social sustainability indicators associated with WFA in the literature.

As shown in Table 4, approaches that include society at large through local purchases, local hiring, and support to local community initiatives recognise that social impact is better understood through community engagement and household indicators than through national indicator estimates. While social footprint creates a framework in which to consider comprehensive monetary measures for a specific product or activity, social water footprint should consider the opportunity cost and benefit to society associated with agricultural water consumption.

**Table 4.** Social sustainability indicators.

| | WFA Social Indicators per Water Footprint Manual | Indicator Description | Extended Social Indicators for WFA in the Literature | Indicator Description |
|---|---|---|---|---|
| Grouped as basic human needs | Domestic water use | Minimum water for drinking, washing, and cooking | Employment | Jobs/m$^3$ (labour intensity in crop production per cubic metre; can include direct and indirect employment) |
| | Minimum allocation of water for food | Secure sufficient level of food supply for all | Income and livelihoods | Effect of water on income in local communities, households, and individuals |
| | Employment | Affected by water-user principle/polluter-pays principle (like basic rules of fairness aspect) | Community health | Water pollution, waterborne diseases, and access to healthcare services |
| | Human health | Water resources for sustainable development in South Africa | Resilience and adaption | Communities' capacity to adapt to changing water availability |
| | Employment | Jobs/m$^3$ | | |
| | Distribution of welfare | Water footprint as an indicator of social, environmental, and economic sustainability | Social accounting | Social accounting method to understand the effect of water consumption on local communities and vulnerable populations |
| | Food security | Simply states that reduced food security is an indicator of secondary impacts | Stakeholder engagement | WFA agents to update and engage all stakeholders involved in water consumption |

**Table 4.** *Cont.*

| | WFA Social Indicators per Water Footprint Manual | Indicator Description | Extended Social Indicators for WFA in the Literature | Indicator Description |
|---|---|---|---|---|
| Grouped as basic rules of fairness | Income | Simply states that reduced income is an indication of secondary impacts | | |
| | Equity | Fair use of public goods such as water (WFA per capita) | | |
| | Water user pays principle/polluter pays principle | | | |

Source: Hoekstra et al. [11]; Altobelli et al. [50].

In addition to the above thematic review on water-footprint-assessment methodologies, a summary of the results of a traditional literature search for sustainability analysis of water for agricultural production is given in Table 5. Table 5 includes a summary of the findings.

**Table 5.** Papers that provide analyses of sustainable water use for agricultural production.

| Author(s) | Objective | Method | Findings | Remarks |
|---|---|---|---|---|
| Gartsiyanova et al. [51] | Assess water quality as a key component in the water-energy-food (WEF) nexus | Canadian Complex Water Quality Index | Authors highlighted water quality, as well as the physiochemical characteristics of water, as having economic, environmental, and social impacts. | The authors did not consider society as a component of water use, but rather identified factors that have a social impact on water use, an approach that leaves unexplored the social aspects of this finite resource. |
| Streimikis and Baležentis [52] | To link rural policy goals with sustainable development in an agricultural sustainability assessment framework | Literature review | The article examined sustainable agricultural development, agricultural-sustainability concepts, and sustainability-evaluation methodologies and tools created for the agricultural industry. | The authors concluded that sustainability issues are country-specific and are based on climate and environmental policy. |
| Velasco-Muñoz [53] | A review of 25 years of international research on sustainable water use in agriculture | Literature review | Only 14.5% of articles on sustainable water use in agriculture used social science approaches, while 70% considered environmental science approaches. | The author report a direct link between sustainability concepts applied to water in agriculture and water-use efficiency and ecosystem concepts, but not society. |

**Table 5.** *Cont.*

| Author(s) | Objective | Method | Findings | Remarks |
|---|---|---|---|---|
| Fabiani et al. [54] | WEF nexus for sustainability assessment at farm level | WEF nexus assessment | The WEF nexus allows the achievement of agricultural sustainability goals by increasing competitiveness and transitioning to environmentally friendly production. | Although the authors mention society as an integral part of sustainable water use, they considered only the environmental sustainability of fertilisation and economic performance as a measure of sustainable water use. |

Summary:

- Water use is referred to as having an economic and a physical component, where the physical is expressed by environmental indicators around water quality.
- A general analysis of sustainability-related studies on water in agriculture confirms that water use is not recognised as an element of society. Despite the repeated appearance of the idea that sustainability contains environmental, economic, and social elements, proposed frameworks on sustainable water use ignore the social element of this resource, and consequently, no attention is paid to the social aspects of water use.
- There is thus a research gap in the available literature with regard to the social element of sustainable water use, which would validate the social aspects of water use.

## 4. Discussion

Water scarcity leaves people in a vulnerable state, and water assessments that address the environmental and economic impacts of such vulnerabilities without considering social consequences perpetuate an injustice against the people affected by such scarcity. According to Adkhamov [55], water is the medium through which climate change manifests its far-reaching consequences for society, as it fulfils three main needs: to replenish the environment, to add value, and to sustain human needs. Water scarcity is therefore not only a physical-biological problem, but also an economic and social problem. Sustainable or unsustainable use of this finite resource encompass economic, environmental and the social benefits or consequences of such use. It is certain that approaches that fail to recognise the social aspects of water use cannot result in an effective understanding of what the sustainable use of this resource entails.

Scientific mapping indicates that LCA and WFA tools are dominant methods for assessing sustainable water use, as is illustrated by the co-authorship linkages in Figure 3. The other common thread was the need to integrate socio-economic water demand within measures of sustainable water use. The network relationships between the keywords used in the scientific-paper selection were dominated by environmental/health indicators. A closer look at the keyword linkages presented in Figure 5a–d showed smaller, less dominant clusters that included social keywords such as "equity", "consumption behavior" and "well-being" without direct linkages to each other.

The thematic review of social indicators for agricultural production revealed society-related indicators like labour-related conditions, as well as factors such as employment, quality of life and decent livelihoods. Social indicators in these papers are grouped into different production-related categories. This review highlights different stakeholders in the production processes and social indicators associated with these stakeholders. It is also clear that there is no standardized method for determining what to consider as a social indicator or how these indicators should be measured. The thematic review on the grey literature includes WFA and LCA among social water-use indicator frameworks review, highlighting GWI's influence on water governance, and we can observe compartmentalized social water-use indicators, as in Section 3.2. Table 2 indicates a lack of consensus on what to consider as a social water indicator and how to measure such an indicator. The individual indicators listed in Table 3 were different from those indicated in the frameworks/tools

given in Section 3.3.1 even though the question asked in this section was similar to that asked in the aforementioned section, only specifying indicators within tools. This result could be an inconsistency associated with the AI tool used for this search.

As seen in Table 5, studies that consider sustainable water use and water characteristics for agricultural use do not consider the social aspect of water use as a direct impact of the use of water for agricultural purposes, but rather consider it as a "by-product" of this water use. This is a dangerous approach considering that at the policy level—whether one considers the AU's Agenda 2063, the UN's SDGs, or South Africa's NDP 2012—a focus on the environmental, economic, and social impact of water is necessary for the sustainable use of water for future generations. Additionally, results from this literature review indicated that a growing body of literature is quantifying and attempting to measure the social element of sustainability. Therefore, this paper highlighted a further need to quantify social sustainability within water-use research because if there is social value associated with agricultural water use, there is possible social devaluation attached to unsustainable freshwater use along agricultural value chains.

Social sustainability focuses on people indicators such as education, skills, experience, consumption, income, and employment, which are globally different and unique, even on a provincial level, and all of which are connected to a condition of vulnerability.

## 5. Conclusions

Defining core topics and associated search terms is a crucial part of any systematic literature search. This paper highlights that the three pillars of sustainability (environment, economy, and society) specified by the UN's Agenda 2030, which are necessary for Africa's Agenda 2063 and the South African NDP (2012), have not been given the same priority in studies of the use of freshwater resources, even though water scarcity poses a significant risk to society (people) and the economy at large.

As seen from the scientific mapping, social sustainability indicators are widespread and are not uniform throughout the literature. Co-occurrence keywords were not specific for social indicators; instead, the literature relies heavily on already established environmental indicators as holistic sustainability indicators. Co-authorship results illustrated that specific tools related to the social element of sustainability are in common use. These tools were found to be the LCA and WFA. These tools involve the social element of sustainability but do not offer sufficient indicators to support measurement of this social element. From the network linkages, it is evident that the integration of social sustainability into assessments of freshwater use is in its infancy and that both methods and indicators to close this literature gap are necessary.

The thematic reviews indicated that literature on social sustainability from broader agricultural value-chain studies and the grey literature complement and enable better understanding of the methods, indicators, and the role that the social element of sustainability for freshwater should play in research on sustainable water use. As shown in Table 4, local community initiatives recognise that social impact is better understood through community engagement and household indicators rather than through national indicator estimates. Social-footprint analysis demonstrated how a comprehensive monetary measure could include these community-based indicators. The WFA's social indices of sustainability include human health, employment, welfare distribution, and food security. Social sustainability focuses on human indicators such as education, skills, experience, consumption, income, and employment, which vary globally and uniquely, even at the provincial level, and all of which are linked to and affected by water availability.

The purpose of this paper was to review the current literature on social sustainability indicators for water use in agriculture. This paper gathered literature on methods, indicators, and measurements for assessing social sustainability, in the form of a systematic review following the PRISMA framework, thematic reviews, and a review of the grey literature to emphasise the gap in scientific literature, and the lack of recognition of the social aspect of water use. Although the social aspect of water use can be expressed in

terms of domestic water supply or in terms of jobs per cubic meter at the farm/processing level, as well as in more than 50 social (people)-related indicators and measurements of the sustainable use of water, the literature does not capture the social value of water, and fails to validate that the social aspect of water use should be a crucial consideration that informs sustainable agricultural water use.

Future research should consider social sustainability valuation of freshwater resources along agricultural value chains. The social aspect of sustainable water use should capture the social value of water incorporated into a product along its value chain. The development of a social water-productivity indicator for WFA would advance research on the impact of freshwater resources in relation to social sustainability in the context of agricultural production and consumption.

**Supplementary Materials:** The following supporting information is included for this review: https://www.mdpi.com/article/10.3390/hydrology11050072/s1, Table S1: Query string from Scopus; Table S2: Inclusion/exclusion criteria for data search

**Author Contributions:** All the authors significantly contributed to this review paper's preparation. P.M.P. was involved in study design, conceptualisation, review, and writing the first draft. H.J. and Y.T.B. aided in the study design, conceptualisation, review, and writing of the final draft. All authors have read and agreed to the published version of the manuscript.

**Funding:** This research received no external funding.

**Conflicts of Interest:** The authors declare no conflicts of interest.

## Appendix A

Table 1 indicates the search query used to gather literature from Scopus.

## Appendix B

Table 2 indicates the inclusion and exclusion criteria for the literature search in Scopus.

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
