# Peer review of "A Systematic Review of Social Sustainability Indicators for Water Use along the Agricultural Value Chain"

_hydrology, doi:10.3390/hydrology11050072_

Round 1
Reviewer 1 Report
Comments and Suggestions for Authors
In general, this manuscript requires major revision and improvement (see comments and suggestions below). The manuscript needs to be rewritten and systematically rearranged to follow a structured format, including Abstract, Introduction, Research Methods, Results and Discussion, and Conclusion.

In general, this manuscript requires major revision and improvement (see comments and suggestions below). The manuscript needs to be rewritten and systematically rearranged to follow a structured format, including Abstract, Introduction, Research Methods, Results and Discussion, and Conclusion.
Author Response
Thank you for the time taken to review this article. Your inputs are highly appreciated.
Reviewer 2 Report
Comments and Suggestions for Authors
This article provides a systematic review of literature of Social Sustainability indicators of water along agricultural value chain. In my opinion, the paper presents significant findings that encourage reflection.
However, in my point of view, the focus is more on sustainability than hydrology. I suggest referring the article to another journal, whose scope is more focused on sustainability.
Author Response
Thank you for taking the time to review this paper. Your inputs are highly appreciated.

Reviewer 3 Report
Comments and Suggestions for Authors
Dear authors,
Thank you for the possibility to review to paper and thereby get a glimpse of the the social sustainability aspects related to water use in agriculture. It highlights the under-researched area of social sustainability within water consumption studies and aims to identify and analyze social sustainability tools, indicators, and measurements across agricultural value chains. The systematic review, grounded in the PRISMA framework and enhanced by scientific mapping through Vosviewer software, thematic reviews, and AI-generated grey literature, underscores the absence of standardized social sustainability indicators in the existing literature. The article ambitiously covers a wide range of literature, drawing from 314 papers over a decade, providing a broad perspective on the topic. Addressing the social aspects of water sustainability in agriculture is both relevant and timely, given the increasing global focus on sustainable resource management and the need for holistic sustainability assessments that include social factors.
I think this article has good potential but before being considered ready for publication, some aspects need to be clarified and improved. However, a few points could have amplified the article's impact:
- While the article's scope is a strength, its wide-ranging nature might also dilute the focus. A more pointed discussion on the implications of identified indicators for policy and practice could enhance its applicability.
- The article presents a wealth of information but could benefit from a more cohesive integration of findings from different sections. Highlighting how these findings interrelate and painting a clearer picture of the current state of research in social sustainability indicators for water use in agriculture could make the narrative more compelling.
- Expanding on the practical implications of the research findings and suggesting specific applications for the indicators identified in agricultural practices or policy-making could make the article more useful for practitioners and policymakers.
- While the article identifies gaps in the literature, it could further elaborate on specific future research directions, possibly suggesting methodologies or frameworks that could be employed to fill these gaps.
- Pleas also make the positioning of the paper clearer already at the final remarks (8). Moreover, I am missing more answers to “so what questions” and implications.
Comments on the Quality of English Language
English language and style are minor spell check required.
Author Response
Thank you for the time taken to review this paper. Your inputs are highly appreciated.

Round 2
Reviewer 1 Report
Comments and Suggestions for Authors
The author has not paid attention to the previous suggestions (yellow marked below), particularly regarding the research design. The necessity of improving the manuscript is crucial to achieve a high-quality systematic literature review. The research design or methods section should contain only a detailed explanation of the methods used. Therefore, the results and discussion should be written separately from the research design. The manuscript should follow the structure as outlined below:
Introduction
Research Design (Methodology)
Results and Discussion
Conclusion
The decision will be made after rearranging the manuscript to the suggested structure. Please provide a response matrix.

minor improvement
Reviewer 2 Report
Comments and Suggestions for Authors
As I said before, this manuscript brings interesting findings.
In my view, authors should improve the organization of their manuscript's sections.
I suggest improving and modifying section 2. Instead "Research Design", rename the section: "Methods" or "methodology" and in this section explain the research design and briefly describe all the methods used, like Scientific Mapping, and the technology (softwares, AI) used. The methods and results are mixed in sections 3, 4, 5, and 6. To make the manuscript more "fluent and readable", it's important to separate the "methods" and "results + discussion), in my opinion.
Please, check Abstract line 13 "Form 314 papers".
The authors didn't cite PRISMA as recommended by Citing PRISMA 2020 — PRISMA statement (prisma-statement.org). Please, verify.
Figure 3 appears to be incomplete. Please, check.
What is the reason for using chatCPT to search for grey literature? I would appreciate it if the authors could provide other options for finding grey literature and explain their decision (chatGPT).
